# Feasibility Study of Reclaimed Asphalt Pavements (RAP) as Recycled Aggregates Used in Rigid Pavement Construction

**DOI:** 10.3390/ma16041504

**Published:** 2023-02-10

**Authors:** M. K. Diptikanta Rout, Surya Kant Sahdeo, Sabyasachi Biswas, Krishanu Roy, Abdhesh Kumar Sinha

**Affiliations:** 1Department of Civil Engineering, National Institute of Technology, Jamshedpur 831014, India; 2Department of Civil Engineering, Thapar Institute of Engineering & Technology, Patiala 147004, India; 3Department of Civil Engineering, National Institute of Technology, Durgapur 713209, India; 4School of Engineering, University of Waikato, Hamilton 3216, New Zealand

**Keywords:** RAP, rigid pavement, silica waste, mechanical properties, sustainability

## Abstract

Reclaimed Asphalt Pavement (RAP) as recycled aggregates is a relatively new construction process of rigid pavements due to the scarcity and degradation of natural aggregates. This study aims at the sequential characterization of RAP aggregate to obtain optimized proportions for strength. For this purpose, RAP aggregates were used for the replacement of natural aggregates (NA) in the concrete mix which was achieved by varying from 0–100%. Furthermore, zirconia silica fume (ZSF) was used as a partial replacement of the cement in the concrete mix, replacing Ordinary Portland Cement (OPC). Experimental studies have shown that the incorporation of washed RAP (WRAP) slightly reduces the compressive strength of concrete by 2.7–37.35% as compared to the reference control concrete mix. Although the 7-days, 28-days and 56-day compressive strength of WRAP recycled aggregate-based concrete is slightly better than the 7-days, 28-days and 56-day compressive strength of dirty RAP (DRAP) recycled aggregate-based concrete. A similar trend was observed in the flexural strength and split tensile strength of WRAP recycled aggregate-based. Overall, the results show that 40% WRAP recycled aggregates with 10% ZSF as a replacement for cement outperform DRAP aggregates in concrete mixes. According to the ANOVA results, the combination of ZSF and WRAP aggregates met the cement concrete pavement strength standard, thereby contributing to sustainable development. Reclaimed Asphalt Concrete Pavements (RACP) are now seen as a potential and long-term answer to the present environmental and economic crisis.

## 1. Introduction

Due to the depletion of limited sources of natural aggregates (NA), there is a scarcity in the global market leading to a surge in the cost of NA. One of the best alternative sustainable approaches is to utilize recycled aggregates in road construction [1,2,3]. Most of the past research reported that construction and demolition (C&D) materials from the extraction of road and building waste can be employed in highway construction to stabilize disposal problems [4,5,6,7]. In India, MORTH (Ministry of Road Transport and Highways) has recently chosen to construct rigid pavement over flexible pavements as the life span is more than 30 years. With this context, Reclaimed Asphalt Pavement (RAP) materials are the better option to use in the road section for the production of cement concrete pavements [8,9,10].

RAP is the residue material collected from the asphalt pavement by the ripping and crushing process [11,12,13]. Figure 1 showed loads of stockpiled RAP have been wastage at the roadside after the milling from the flexible pavement. In India, an enormous amount of RAP materials are produced as waste every year at the plant site [14,15,16]. On the other hand, approximately 835 kg of CO_2_ is released per ton of cement produced, which amounts to 7–9% of total carbon dioxide emissions [17]. So, recycling concrete pavement materials is necessary to implement the idea of sustainability in the transportation industry [18,19,20]. RAP is being widely utilized for various pavement applications such as wearing courses, bases or sub-bases, and dry lean concrete bases in global trends [21,22,23]. Brand and Roesler [24] studied the effect of the partial replacement of RAP aggregates by natural aggregates increased the workability but reduced the hardened properties of concrete mixes. Singh et al. [25] cleared that WRAP recycled aggregates processed from washing and abrasion aggregate techniques provide better workability than dirty RAP recycled aggregates (DRAP). In this regard, the RAP recycled aggregate was screened into different methods to segregate it into DRAP and WRAP aggregates. DRAP recycled aggregates mean that the presence of dust particles around the RAP recycled aggregates causes the reduction in mechanical properties of concrete whereas washing the aggregates before the incorporation mix enhances the fresh properties of concrete termed WRAP [26]. In order to create a window and strengthen the interaction between the RAP and cement mortar, washing the RAP may have minimized the asphalt coating layer that was surrounding the aggregates [27]. For the consideration of strength criteria, most researchers indicated that the compressive strength of RAP concrete mixes decreases with the comparison of natural aggregate [28,29,30]. Moreover, the presence of agglomerated particles leads to a drastic reduction in compressive strength [31]. Similarly, the flexural and split tensile strength of RAP mixes decreases with increases in RAP aggregates in cement concrete production [30,31,32,33]. The weak interface between the cement mortar and RAP aggregates was blamed for the deterioration of RAP characteristics inclusive in cement concrete mixes as well as the adequate adhesion of asphalt film in cement paste forms the larger number of agglomerated particles in RAP [34,35,36,37,38]. It is evident from the micro-structure studies, the formation of a poor interfacial zone (ITZ) between the RAP and cement mortar also causes the reduction of concrete strength in rigid pavement [32,39]. The utilization of solid waste materials collected from industry known as silica fume (SF) or Nano-silica (NS) might greater enhance the mechanical and durability properties of standard concrete significantly [40,41]. Henceforth, the utilization of 100% RAP recycled aggregates is typically not advised due to their subpar performance when it is compared to either natural coarse or fine aggregates in concrete mixes [42]. So, several researchers recommended 40–50% of replacement of RAP aggregates in concrete mixes to achieve the benchmark of strength for the production of RAP concrete [37,38,39,40,41,42,43,44,45,46,47]. With the considerations of statistical analysis, mathematical modelling gives a high confidence level to predict the strength of concrete. In contrast, Sahdeo et al. [48] validated the experimental results on the different fractions of RAP concrete mixes used in previous concrete pavements by the statistical model analysis whereas, Dubey et al. [49] found that the moisture content is less susceptible than the fine RAP percentages affecting the strength characteristics of dry lean concrete (DLC) mixes by Two-way analysis of variance (ANOVA). Moreover, Saboo et al. [50] indicated that the effect of fly ash content dominated the permeability and compressive strength of the previous concrete pavement. So, the ANOVA models can be used along with the experimental results to determine the significance of the variables for the concrete mix with the incorporation of the pozzolanic materials.

Therefore, efforts have been made to properly characterize collected RAP materials from different sources of local sites. In this current study, we develop the strength and potential of RAP aggregates by removing dust, agglomeration, and coated asphalt. The literature that is now available claims the inclusion of RAP degrades the concrete characteristics but nonetheless appears to be an innovative way to partially substitute virgin aggregates. Likewise, this article also addressed the inclusion of locally accessible materials, which will aid in the construction of sustainable rigid pavement and a greener environment.

## 2. Research Objectives

Most of the literature pertaining to RAP recycled aggregates in cement concrete mixes has either determined the optimal proportions of RAP based on strength criteria or characterized the RAP based on physical and chemical properties. There is limited research that addresses the properties of RAP recycled aggregates as well as the optimal proportions of RAP used in concrete mixes. Consequently, an attempt has been made in the present study to identify the physical properties of RAP recycled aggregates and to strengthen the mechanical properties of concrete mixtures incorporating RAP recycled aggregates as replacements for natural aggregates and incorporating various amounts of ZSF as industrial waste. In this regard, in the present study we also establish whether the specified aggregates were coarse or fine, the gradation curves of NA, DRAP, and WRAP recycled aggregates have been drawn. Further, few researchers have proposed that concrete pavement construction should use no more than 50% RAP in the surface course. Using literature as a basis, efforts have been made to enhance the use of WRAP aggregates in rigid pavement. Furthermore, this is a technique to improve the strength qualities of RAP recycled aggregates by using locally accessible industrial wastes and developing the co-relationship between them. For varied RAP percentages, the durability properties such as water absorption and sorptivity have also been examined. A field emission Scanning Electron Microscopy (Fe-SEM) technique was also used to further understand the micro-structure behaviour of RAP recycled aggregates and cement mortar matrix. Therefore, the objective of this laboratory investigation is to determine the optimum feasible RAP inclusion level for the manufacturing of concrete mixes based on various factors for concrete pavements in presence of ZSF.

In the present study, the effect of two independent variables on a continuous dependent variable was determined by two-way ANOVA at a 5% significant level (α). In this process, the ANOVA technique was performed on Density, 28-days compressive strength, 28-days Flexural strength, and 28-days split tensile strengths to examine the significant differences between WRAP recycled aggregate-based concrete mix, DRAP recycled aggregate-based concrete mix, and reference concrete mix.

## 3. Physical Characterization of RAP

Most researchers have reported that the RAP recycled aggregates are comparable to the natural aggregates so this can be used for rigid pavement construction [8,25,35,37,51]. Roesler et al. [28] fractionated the RAP aggregates on a 4.75 mm sieve to separate coarse and finer fractions whereas Zhu et al. [52] considered 2.36 mm sieve passing as fine RAP recycled aggregates from the characterization technique. Masi et al. [53] characterized the RAP recycled aggregate as having greater than 4 mm as coarse RAP (CR) fraction and less than the same size as fine RAP (FR). A fine aggregate to be used in the production of cement concrete mixes should contain a minimum of 35% finer fraction passing through 600 microns IS sieve for gradation perspective [54]. Similarly, ASTM (American Society for Testing and Materials, 2016) recommended that a minimum of 25% finer particles be used as fine aggregates in cement concrete mixes.

The RAP is classified into different ways of characterization in terms of washed RAP, dirty RAP, Abrasion and Attrition treated RAP, surface treated RAP, dust RAP, smooth textured RAP, and contaminated RAP [8,25,55,56]. The specific gravity of RAP usually ranges between 2.26–2.63 due to the presence of low-density asphalt coating [11,28,57]. But in this present study, the specific gravity of DRAP is slightly greater than the WRAP aggregates. Meanwhile, the water absorption of RAP is usually lower than the natural aggregates due to its dust contaminants' presence in asphalt coating [58,59,60]. The accumulation of dust layers around the periphery of aggregates is the primary cause of higher water absorption in DRAP aggregates. The aggregate impact value of DRAP and WRAP aggregates are lesser than the natural aggregates as it indicates the lesser toughness properties of the recycled aggregates whereas Los-Angeles abrasion value of WRAP aggregates was lesser than the DRAP due to the removal of the loose dust layer. The combined flakiness and elongation indices for DRAP and WRAP aggregates were found to be 28.56% and 26.91% respectively. The details of the physical characterization of DRAP and WRAP aggregates are compared to the natural aggregates in Table 1.

## 4. Experimental Investigation

### 4.1. Materials

Ordinary Portland Cement (OPC) with a specific gravity of 3.15 as per IS: 12269-2013 was used in this current study [61]. The initial and final setting times were recorded as 60 and 280 min, respectively. The consistency was 31.5%, and the soundness was 2.8 also noted. Twenty years old RAP recycled aggregates were procured from a different local site in Jharkhand whose specifications are listed in Table 2. RAP recycled aggregates were obtained from the sample using the ripping and crushing method and its characterization is shown in Figure 2. WRAP recycled aggregate is obtained after the washing and cleaning process of normal raw RAP aggregate obtained from the site. Whereas, an unclean membrane with asphalt-coated recycled RAP aggregate is known as DRAP.

Concrete mixtures containing varying amounts of locally available industrial waste i.e., Zirconia silica fumes (ZSF) were added in different proportions. The considered admixtures, whose specific gravity and surface area were 2.3 and 18,500 cm^2^/g respectively (mentioned in Table 1). Figure 3 depicts grain size distribution curves of virgin, DRAP, and WRAP aggregates in compliance with IS: 383-1970 [62]. The grading curve represents the cumulative percentages of sieve passage as ordinated to the logarithm scale as abscissa. The ZSF admixture is derived from Zirconia compounds by a fusion process in the industry having fine granularity features. It is a powder form of white colour in nature and characterized by a particle size of less than 1 micron for the present study. Potable water confirming to IS: 456-2000 [63] was used for the preparation of concrete mixes. The chemical compositions of cement and ZSF admixtures are tabulated in Table 1 from the X-ray fluorescence (XRF) analysis.

### 4.2. Mix Design

The cement concrete guidelines as per Indian Road Congress (IRC): 44-2017 criteria are followed for pavement mix design [64]. Concrete mixes were designed with a water content (*w*/*c*) ratio of 0.46 and naphthalene-based super-plasticizers 0.5% by weight of cement. Table 3 shows the different mix proportions of the six mixture contents. The study compares the casting and testing of M40 concrete mixes using DRAP and WRAP recycled aggregates in place of various natural coarse aggregates (NCA) and natural fine aggregates (NFA). In the present study, some percentage, of course, RAP aggregates partially replace the natural coarse aggregate for preparing the concrete mix. In this sense, 360 kg/m^3^ OPC has been used in the control concrete mix as a reference mix. Also, 0–30% cement weight equivalent of ZSF has been used in the concrete mix which has been considered as a partial replacement for cement.

## 5. Testing Procedure

For this experiment, the physical and mechanical properties of used aggregates including their specific gravity, water absorption, aggregate impact, Los angles abrasion, crushing, and density was tested in compliance with IS: 2386 [65]. To access the workability of considered mixtures, the slump test cone method was used in the laboratory with the ASTMC143 guidelines [66]. The compressive strength of concrete mixes was evaluated by preparing a specimen of standard cube size 150 mm dimensions in accordance with IS: 516 [67]. The cube was cured by submerging it in water for a period of 7, 28, and 56 days of curing. After the given period of curing, the specimen was tested by a compression machine. The compressive strength fc was calculated using Equation (1)
(1)$$fc=\frac{P}{A}$$

Flexural strength $\left(\mathrm{fr}\right)$ was measured by 100 mm × 100 mm × 500 mm beam for the same days of curing as per IS: 516 [67] using Equation (2). Similarly, the split tensile strength (ft) was measured with a cylindrical specimen having dimensions: 10 cm in diameter (D) and 20 cm in length (L) in accordance with IS: 5816 [68] using Equation (3)
(2)$$\mathrm{fr}=\frac{\mathrm{PL}}{{\mathrm{bd}}^{2}}$$
(3)$$\mathrm{ft}=\frac{2P}{\pi \mathrm{DL}}$$
where $\left(\mathrm{ft}\right)$ is expressed as Mpa, P = maximum load is applied (kN), A is the cross-sectional area, D is in cylindrical dimension in mm, and L = length in mm. The water absorption was measured by cubical specimens of size 100 mm × 100 mm at 28 and 56 days of moist curing period in accordance with ASTM C 642 [69]. The difference between saturated surface-dry (SSD) and oven-dry (OD) mass was assumed to represent as water absorption of the concrete mixes. The sorptivity (S) is the degree of water absorption through capillary action was determined by Hall’s principle using Equation (4)
(4)I=S ×√t 
where I = capillary water ingress in mm and t = time in min. The influence of mineral admixture on the microstructure of cured concrete as well as the ITZ produced between RAP aggregates and cement mortar was examined by Scanning Electron Microscopes (SEM) images. Figure 4 depicts the procurement and test process of RAP recycled aggregates used in this study.

## 6. Result and Discussions

### 6.1. Influence of ZSF in the Mechanical Properties of the Cement Conctre

Normally addition of silica fume (SF) to the concrete mixtures is commonly improved the mechanical properties of the concrete mix [70,71]. Zirconia silica fume (ZSF) is a shapeless, chemically active byproduct from the silicon or ferrosilicon industry. According to previous findings [70], the strength provided by adding SF as an admixture to the concrete mix was fairly the same as the strength of another concrete mix, when SF was included as a replacement for cement [72,73]. Therefore, in the present study, ZSF has been used as partially replacing ordinary cement in the concrete mix.

In the present study, cubes, cylinders, and beam specimens were cast and kept in square shaped curing tank in the laboratory for the specified 7 and 28 days of curing ages. The result of the 7 and 28 days compressive and split tensile strength are shown in Table 4.

According to the experimental results, the addition of ZSF to the concrete mix as a partial replacement of cement increased the compressive strength of the concrete. These results reveal similar trends that are consistent with all previous studies [70,74,75]. A positive increase in compressive strength was observed which was more noticeable in standard concrete mix (i.e., reference concrete mix) as compared to the concrete mix containing ZSF. The results of this test show that incorporation of ZSF in the range of 5–10% as cement replacement increased the 7-day compressive strength of the concrete mix by 7.1–10.5% compared to the 7-day compressive strength of the reference concrete mix. Similarly, the 28-day compressive strength increases by 7.5–14.17% with the incorporation of ZSF in the 5–10% range. The main influence of ZSF on the development of the compressive strength of concrete occurs initial stage (between 7 to 28 days). Though, incorporation of ZSF in the range of 15–30% by weight of cement does little or no significant help in improving the compressive strength of concrete. Interestingly, the results show that the 7-day compressive strength marginal increases by 3–6.1% and the 28-day compressive strength marginal increases by 1.2–3.5% (with respect to the compressive strength of the CMSF10 concrete mix) after incorporating 15–30% of ZSF in the concrete mix.

On the other hand, the 7-day split tensile strength increases with a gradual decrementing rate in the range of 3–5.3% after incorporating ZSF in the range of 5–10% in the concrete mix. A similar trend was observed in the 28-day split tensile strength as reported in Table 4. However, there was no significant change in the 7-day and 28-day split tensile strength after incorporating 15–30% of ZSF in the concrete mix from that of the CMSF10 concrete mix. In this sense, the pozzolanic activity of additional ZSF further strengthens the initial compressive strength and split tensile strength of the concrete mixes.

### 6.2. Variation of Compressive Strength in Presence of DRAP and WRAP Aggregates

The cube, cylinder, and beam specimens as prescribed the dimension earlier were cast and kept in square shaped curing tank in the laboratory for the specified 7, 28, and 56 days of curing ages. The compressive strength has achieved the target strength of control mixes at 28 days. Figure 5 shows the compressive strength of concrete in the presence of DRAP or WRAP recycled aggregate as a partial replacement of natural aggregate with 10% of ZSF as a partial replacement of cement.

In this sense, the addition of ZSF helps to fortify the compressive strength of various concrete mixes which can negate the effect on compressive strength caused by the incorporation of DRAP or WRAP recycled aggregate in these concrete mixes.

The 7-day compressive strength was found 32.38 Mpa and 31.34 MPa for WRAP20 and WRAP40 mixes, whereas, WRAP60, WRAP80, and WRAP100 mixes reduced the strength value nearly equal to 36.8%, 36.7%, 31.6%, 2.9% and 19.3%, respectively. However, 28-day compressive strength was 48.76 Mpa and 48.4 MPa for WRAP20 and WRAP40 mixes respectively. Interestingly, a marginal difference has been observed in 28-day and 56-day compressive strength for the concrete mixes WRAP20 and WRAP40. Further addition of the WRAP recycled aggregate proportion in the concrete mixes, decreases the 28-day and 56-day compressive strength of the concrete mixes WRAP60, WRAP80, and WRAP100.

A similar trend was observed in the 28-day and 56-day compressive strength of the concrete, when DRAP recycled aggregates were present in the concrete mix as a partial replacement of NA. In this regard, 7-day compressive strength was found to be 20.45, 19.85, 17.22, 16.89 and 15.48 MPa respectively for DRAP20, DRAP40, DRAP60, DRAP80, and DRAP100 concrete mixes respectively. A similar trend was observed in the 28-day and 56-day compressive strength of the DRAP20, DRAP40, DRAP60, DRAP80, and DRAP100 concrete mixes. On the other hand, the 7-day, 28-day and 56-day compressive strengths of DRAP recycled aggregate-based concrete mixes were lower than those of concrete mixes when the same proportion of WRAP was added to the mixes as a substitute for NA. It was recorded that 36.8% in DRAP20, 36.7% in DRAP40, 31.6% in DRAP60, 22.9% in DRAP80 and 19.3% in DRAP100 showed lower 28-day compressive strength compared to WRAP recycled aggregate-based concrete mixes (i.e., WRAP20, WRAP40, WRAP60, WRAP80, WRAP100). Similarly, a slight decrease in the 28-day and 56-day flexural strength of concrete was observed where WRAP and DRAP recycled aggregates were incorporated in the concrete mix at different proportions (reduction in the strength 34.04%, 37.35%, 24.95%, 9.30% and 18.16% for WRAP20 to DRAP20, WRAP40 to DRAP40, WRAP60 to DRAP60, WRAP80 to DRAP80, WRAP100 to DRAP100 respectively).

Moreover, when OPC was replaced by ZSF, the compressive strength increased at the level of DRAP20, DRAP40, WRAP20 and WRAP40 type of concrete mixes at 28 and 56 days of curing. This improvement indicated the higher concentration of silica content present (i.e., the inclusion of ZSF) in the cement concrete mixes. It has been observed that the addition of DRAP recycled aggregate or WRAP recycled aggregate as replacement of NA in concrete mix reduces the mechanical properties of concrete. As a result, the compressive strength increased as a result of the considerable amount of activated silica fume reacting with portlandite to create calcium silicate hydrated gel (C-S-H) in the hydrated sample. So, it is recommended that 10% ZSF with WRAP recycled aggregates can be used in the surface course of rigid pavements for economical and sustainable construction.

### 6.3. Variation of Flexural Strength in Presence of DRAP and WRAP Aggregates

Similarly to the compressive strength of concrete, the flexural strength of concrete decreased compared to the flexural strength of the reference concrete control mix after the inclusion of WRAP or DRAP. Figure 6 shows that the reduction (in terms of percentage) in flexural strength at 7-days and 28-days was greater than that at 56-days. In this regard, 7-days of flexural strength was found to be 3.9, 3.8, 3.6, 3.0, 2.8, 2.8, 2.6, 2.7, 2.1 and 2.0 MPa respectively for WRAP20, WRAP40, WRAP60, WRAP80, WRAP100, DRAP20, DRAP40, DRAP60, DRAP80 and DRAP100 concrete mixes respectively. It was observed that 28.2% in DRAP20, 31.8% in DRAP40, 25.0% in DRAP60, 30.0% in DRAP80 and 28.6% in DRAP100 showed low decreases in 7-day flexural strength as compared to WRAP recycled aggregate based concrete mixes (i.e., WRAP20, WRAP40, WRAP60, WRAP80, WRAP100). However, a slight reduction in 28-day flexural strength was observed at 14.3%, 13.3%, 12.9%, 22.1% and 22.9% for WRAP20 to DRAP20, WRAP40 to DRAP40, WRAP60 to DRAP60, WRAP80 to DRAP80, WRAP100 to DRAP100 respectively and decrease in the 56-days flexural strength were observed 14.0%, 22.7%, 17.9%, 23.3% and 13.0% for WRAP20 to DRAP20, WRAP40 to DRAP40, WRAP60 to DRAP60, WRAP80 to DRAP80, WRAP100 to DRAP100 respectively. A drop in flexural strength was recorded for the DRAP100 mix due to the presence of loose asphalt layers. In this regard, the use of WRAP recycled aggregates in concrete mixes develops better flexural strength. Additional inclusion of silica fume in the form of ZSF as a replacement for the same amount of cement helps to nullify the specific effect of WRAP in this concrete mix.

### 6.4. Variation of Split Tensile Strength in Presence of DRAP and WRAP Aggregates

The results of split tensile strength at 7-days, 28-days and 56-days are depicted in Figure 7. Similarly to the split tensile of concrete, the split tensile of concrete decreased compared to the split tensile of the reference concrete control mix after the inclusion of WRAP or DRAP. In this regard, 7-days of Split tensile strength was found to be 3.01, 3.14, 2.65, 2.40, 2.20, 2.7, 2.75, 2.59, 2.14 and 2.07 MPa respectively for WRAP20, WRAP40, WRAP60, WRAP80, WRAP100, DRAP20, DRAP40, DRAP60, DRAP80 and DRAP100 concrete mixes respectively. Similarly, it was observed that 0.7% in DRAP20, 8.0% in DRAP40, 5.7% in DRAP60, 5.0% in DRAP80 and 3.2% in DRAP100 showed low decreases in 28-day Split tensile strength as compared to WRAP recycled aggregate based concrete mixes (i.e., WRAP20, WRAP40, WRAP60, WRAP80, WRAP100). However, a slightly greater reduction in 56-day split tensile strength was observed at 14.3%, 13.3%, 12.9%, 22.1% and 22.9% for DRAP20 to DRAP20, for WRAP40 to DRAP40, for WRAP60 to WRAP60, for WRAP80 to DRAP80 and WRAP100 to DRAP100 for respectively. In this regard, the use of WRAP recycled aggregates in concrete mixes develops better flexural strength. Therefore, results show that WRAP aggregates were superior to those from DRAP aggregates in terms of split tensile strength.

### 6.5. Variation of Water Absorption in Presence of WRAP and DRAP Aggregate

Water absorption of various concrete mixes at 28 and 56 days are illustrated in Figure 8. The rate of water absorption decreases with the age of concrete and this pattern is almost constant for all mixes. Water absorption capacity decreases with the incorporation of DRAP and WRAP recycled aggregates in the concrete mix compared to the RM mix. Interestingly, it was observed that 1.9% in RAP20, 13.3% in WRAP40, 64.3% in WRAP60, 29.2 in WRAP80 and 50% in WRAP100 showed decreases in 28-day absorption as compared to WRAP recycled aggregate-based concrete mixes (i.e.*,* DRAP20, DRAP40, DRAP60, DRAP80, DRAP100). Similarly, increases in the 56-days water absorption were observed at 54.5%, 43.2%, 49.0%, 47.8%, and 63.3% for DRAP20 to WRAP20, DRAP40 to WRAP40, DRAP60 to WRAP60, DRAP80 to WRAP80, DRAP100 to WRAP100 respectively. Interestingly, the 56-day Water absorption was lower than the 28-day Water absorption of most concrete mixes.

The present study found that the asphalt layer present in DRAP recycled aggregate which covers the dust layer surrounding the aggregate can primarily be attributed to the higher water absorption rate. Whereas, the addition of DRAP recycled aggregate to the concrete mix increases the water absorption capacity of that mix compared to WRAP-added concrete mix. This behaviour is consistent with the previous research [33,34,35,36].

### 6.6. Variation of Sorptivity Characteristics in Presence of WRAP and DRAP Aggregate

The sorptivity represents the variation of capillary water absorption of the concrete specimens with time√t. The first stage of sorptivity was measured for the initial rate of water absorption and the second stage was calculated for the co-efficient of sorptivity. Figure 9 shows the sorptivity measurements for various RAP mixtures at 28 and 56 days. It was observed that the 28-days co-efficient of sorptivity for WRAP20, WRAP40, WRAP40, WRAP80 and WRAP100 mixtures decreased by 1.9%, 13.3%, 64.3%, 29.2% and 50% from DRAP40, DRAP60, DRAP80 and DRAP100 mixtures, respectively. A similar pattern was found for the 56-day co-efficient of sorptivity of different mixes with WRAP and DRAP. Similar to water absorption, the 56-day coefficient of water sorptivity was lower than the 28-day coefficient of sorptivity of various concrete mixes.

Similarly, the co-efficient of water absorption gradually decreased in the case of similar WRAP recycled aggregates at 56 days of curing ages. However, the values of sorptivity co-efficient lead to higher water absorption for DRAP recycled aggregates at both curing ages (28 and 56 days). This phenomenon might be caused by the infilling of capillary holes by the liquefied asphalt and the water-soaking nature of considered admixtures. Moreover, these findings show that the sorptivity coefficient is higher in the DRAP recycled aggregates as compared to the WRAP recycled aggregates. This attribute may be caused in the form of the water-absorbed phenomenon due to capillary action. So, the analysis suggested from the current investigation that the WRAP recycled aggregates including the optimum content of 10% ZSF are suitable for concrete pavements and improving the strength characteristics.

## 7. Microstructure Observations

The microstructure of concrete mixes signifies the visualization of the morphological changes, pore structure, and interfacial transition zone (ITZ) between cement paste and aggregates in pavement structures [76,77,78]. Past researchers recorded that the asphalt content encircling RAP has been ascribed to the inferior bonding between RAP and mortar matrix [79,80,81,82].

Field-Emission Scanning Electron Microscopic (FE-SEM) images for RAP-based concrete with the mineral admixture ZSF as the partial substitution of OPC are shown in Figure 10. The interfacial transition zone (ITZ) has developed between the RAP particles and cement matrix, as observed. However, SEM images clarified that the ITZ among the RAP and cement composites will have a thicker and denser grade. The presence of voids and ettringite in mixtures was discovered to be detrimental to the structure in acidic conditions. The C-S-H gel formed by the interaction of portlandite (CH) with ZSF hydration at a lower concentration. The partial substitution of OPC by the mineral additives ZSF, was found to minimize the CH concentration and voids that indicated the formation of C-S-H gel. Due to the larger void in the RAP concrete mixtures, a thin micro hair fracture was found on the surface of the cement matrix. The uses of ZSF were discovered to lower portlandite concentration and generate C-H-S gel in the concrete mixtures. Furthermore, the inclusion of mineral components in the mixtures occupies the voids.

## 8. Statistical Analysis

The two-way analysis of variance (ANOVA) normally scrutinizes the influence of two independent variables on their outcome [74,75,83,84,85]. In the present study, the effect of two independent variables (i.e., WRAP40 concrete mix and DRAP40 concrete mix) on a continuous dependent variable (such as Density, 28-days compressive, 28-days Flexural strength and 28-days split tensile strengths) has been determined by two-way ANOVA test at 5% significant level (α). In this process, the ANOVA technique was performed to examine the statistically significant differences between WRAP concrete mix, DRAP based concrete mix based on two hypotheses. In this process, the null hypothesis (H0) specifies that the means of Density/28-days compressive strength/28-days flexural/28-days split tensile strength of two different concrete mixes, i.e., WRAP and DRAP are statistically the same or the alternative hypothesis (H1) specifies that the outcomes are statistically different from each other. Table 5 presented the results obtained from the ANOVA test.

Broadly, the F-value obtained from the ANOVA test is greater than the F-critical and the *p-*value is less than 0.05 at the significance level is 5% indicating that the null hypothesis is rejected. In the present study, the null hypothesis is rejected in all cases, meaning all the resulting parameters of WRAP recycled aggregate-based concrete mixes are significantly different from the resulting parameters of DRAP recycled aggregate-based concrete mixes.

## 9. Conclusions

The current study outlines the possible way to utilize WRAP or DRAP recycled aggregates in rigid pavement construction and their feasible solution for future generations. The WRAP aggregates are the beneficiary for road construction as well as the sustainable alternative to natural aggregates. Some of the concluded remarks from the present study are as follows:Studies have shown that the addition of ZSF lead can improve the compressive strength of various concrete mixes which can negate the effect on compressive strength due to the incorporation of DRAP or WRAP recycled aggregate in these concrete mixes up to a certain percentage.The study also shows that up to 40% of the addition of WRAP in presence of 10% ZSF marginal change in 7-days compressive strength. WRAP60, WRAP80, and WRAP100 mixes reduced the strength value are nearly equal to 36.8%, 36.7%, 31.6%, 2.9% and 19.3% respectively with respect to the RM. Similar behaviour was observed in the 28-day and 56-day compressive strength and flexural strength of the concrete mixes.As result shows that the 7-day and 28-day split tensile strength increases with a rate in the range of 3–5.3% and 4.1–7% after incorporating ZSF in the range of 5–10% in the concrete mix. Nevertheless, there was no significant change in the 7-day and 28-day split tensile strength after incorporating 15–30% of ZSF into the concrete mix (CMSF10 concrete mix i.e., more than 10% of ZSF). This behaviour occurred due to the pozzolanic activity in presence of ZSF during initial the strength development phase.As the different proportions of RAP content increase, the strength of concrete tends to decrease. Interestingly, mechanical properties such as compressive strength and flexural strength increase at the 40% RAP level and reach the benchmark. Moreover, the strength was slightly reduced in the case of tensile strength of RAP aggregates-based concrete.In general, the rate of water absorption decreases within 28- and 56-days of most types of concrete mixes. This decrease (10% to 63%) takes place with the incorporation of DRAP and WRAP recycled aggregates in the concrete mix compared to the Reference concrete mix.The use of WRAP recycled aggregates in concrete mixes was proven to be advantageous in durability aspects in terms of water absorption and sorptivity characteristics of concrete.When WRAP aggregates were substituted with NCA, it was observed that the water absorption and sorptivity characteristics were found to be reduced. The present studies also recommend replacing OPC with 10% ZSF mineral admixtures found to enhance the durability characteristics of the RAP40 mix.Furthermore, it can be concluded from this current study that the partial replacement of cement by ZSF helps in improving the micro-structural bond strength which ultimately increases the value of compressive strength.It is intuitive that the incorporation of mineral additives (ZSF) would contribute to improving the strength of concrete mixes, owing to the densification of the ITZ between RAP aggregates and mortar matrix. However, the extra C-S-H gels could not greatly affect the strength properties. A very small amount of ettringite was found in considered mixtures showing the concrete pavement does not cause an adverse effect.The ANOVA model gives the right way on the different parameters of concrete strength and the results clearly indicate the replacement of ZSF in place of cement to produce higher or similar strength of RAP concrete at the optimal level. This confidence brought a greater idea to use ZSF or other by-products in RAP concrete mixes for the construction of the rigid road.This present laboratory investigation found that asphalt coating should be removed for better aggregate-mortar interaction in concrete mixes and DRAP aggregates and has needed some suitable treatments as compared to the WRAP aggregates.Consumption of RAP recycled aggregates in cement concrete pavement mixes would help minimize the utilization of natural aggregates as well as the cost of RAP concrete pavement. Henceforth, the recycling and reusing of aggregates could environmentally sustainable in terms of eliminating waste disposal issues, improving road aesthetics, reduction of carbon footprint, and being economically viable.So, it is recommended that 10% ZSF with 40% WRAP recycled aggregates can be used in the surface course of rigid pavements for economical and sustainable construction.

## Figures and Tables

**Figure 1 materials-16-01504-f001:**
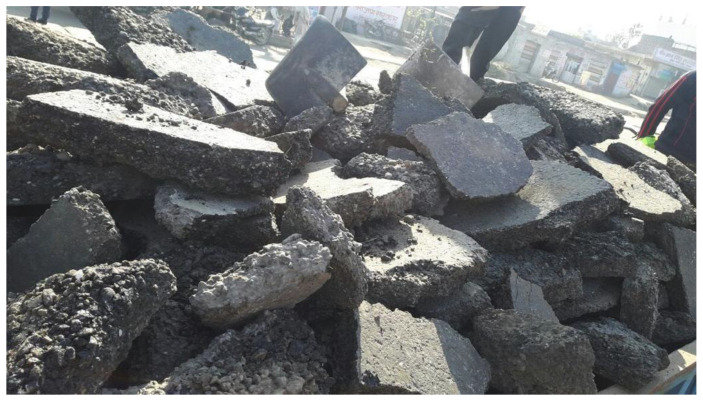
Chunks of RAP stockpiled.

**Figure 2 materials-16-01504-f002:**
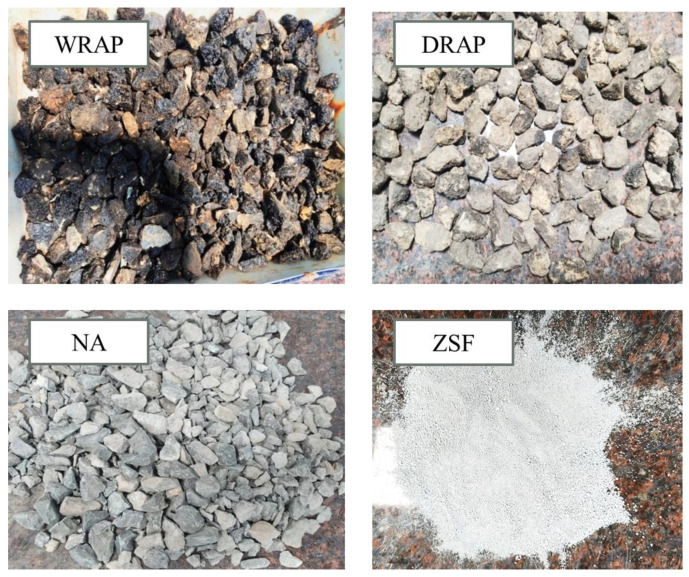
Collection of DRAP, WRAP, NA aggregates and ZSF admixtures used in the present study.

**Figure 3 materials-16-01504-f003:**
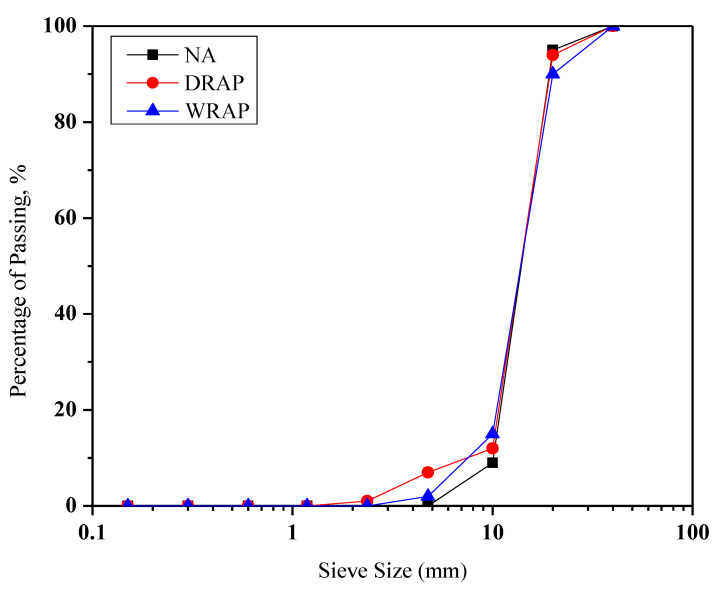
Particle size Distribution analysis of NA, DRAP and WRAP aggregates.

**Figure 4 materials-16-01504-f004:**
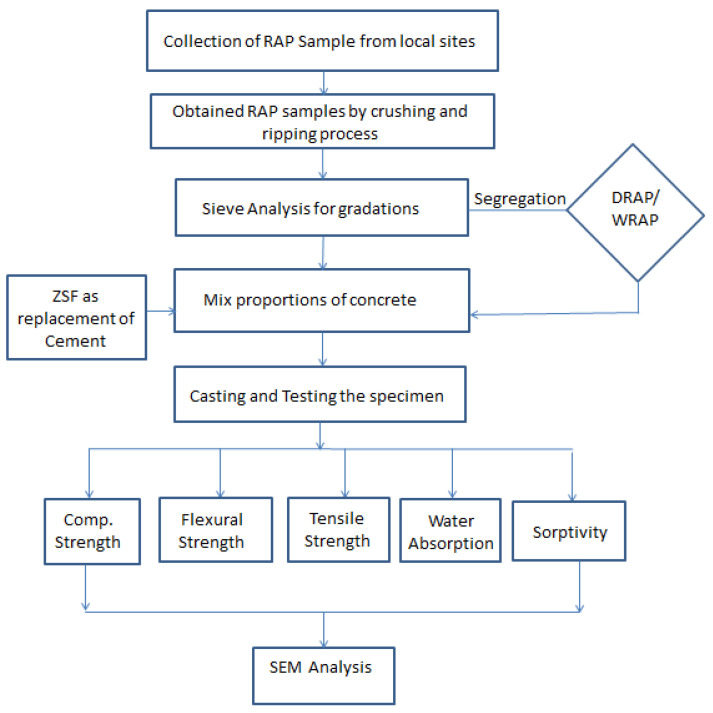
Flowchart for the testing procedure.

**Figure 5 materials-16-01504-f005:**
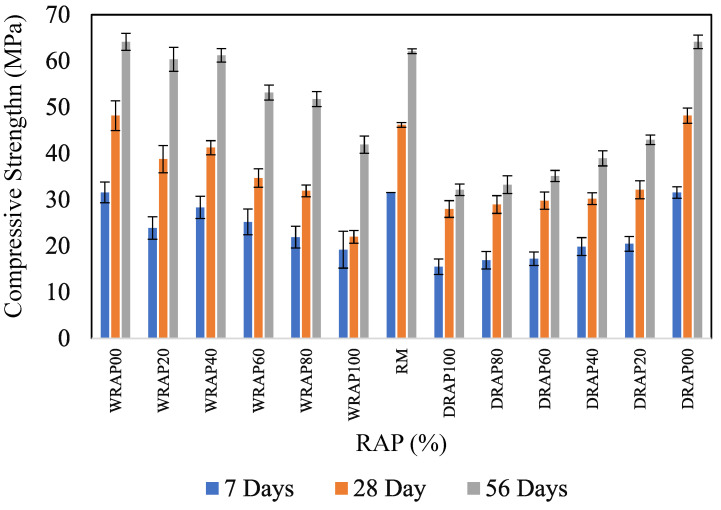
Influence of Compressive Strength in DRAP and WRAP-based Concrete.

**Figure 6 materials-16-01504-f006:**
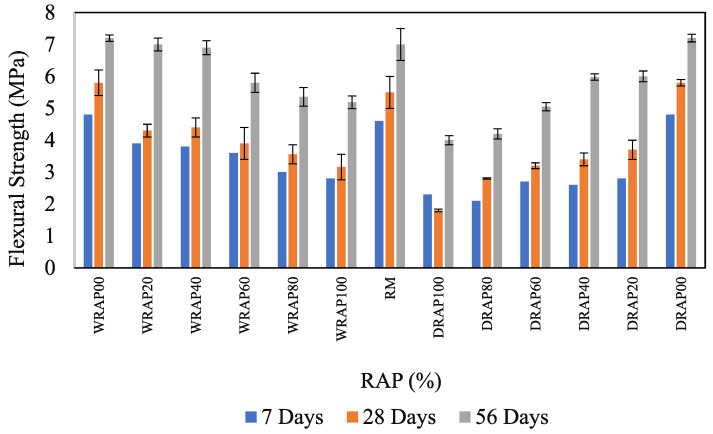
Influence of flexural strength in RAP-based concrete.

**Figure 7 materials-16-01504-f007:**
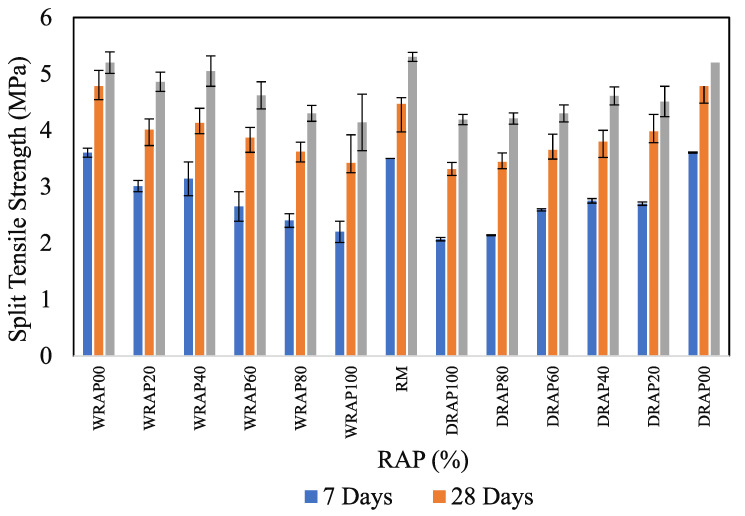
Influence of Split tensile strength in RAP-based concrete.

**Figure 8 materials-16-01504-f008:**
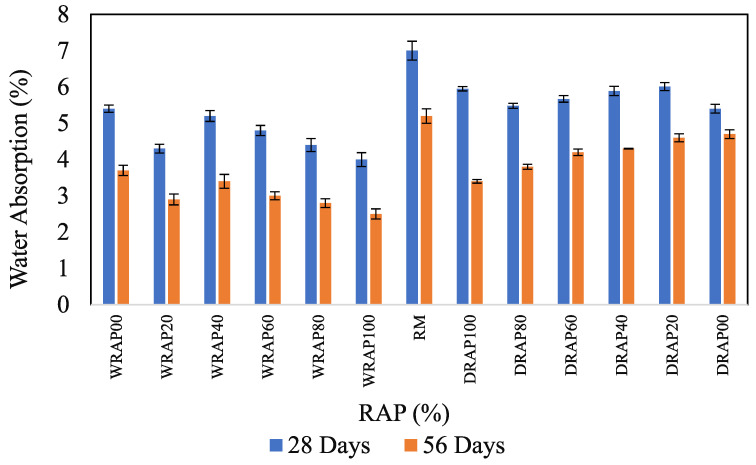
Water absorption characteristics of RAP concrete.

**Figure 9 materials-16-01504-f009:**
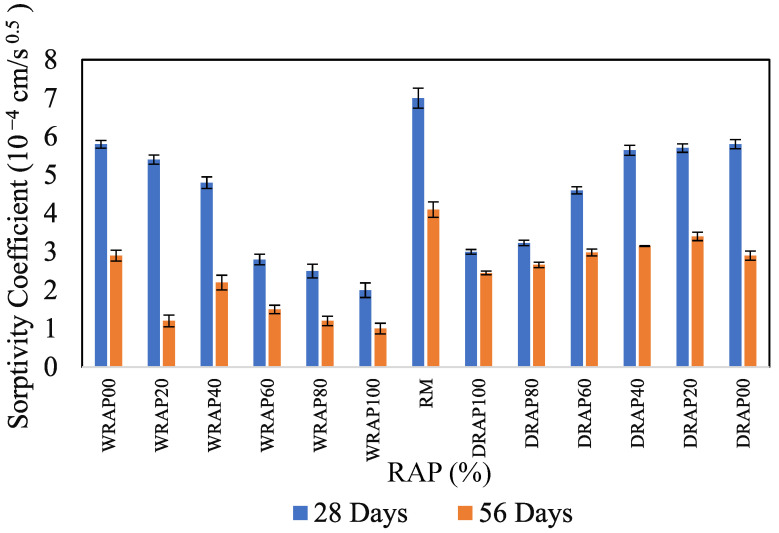
Variations of sorptivity coefficient on RAP concrete mixes.

**Figure 10 materials-16-01504-f010:**
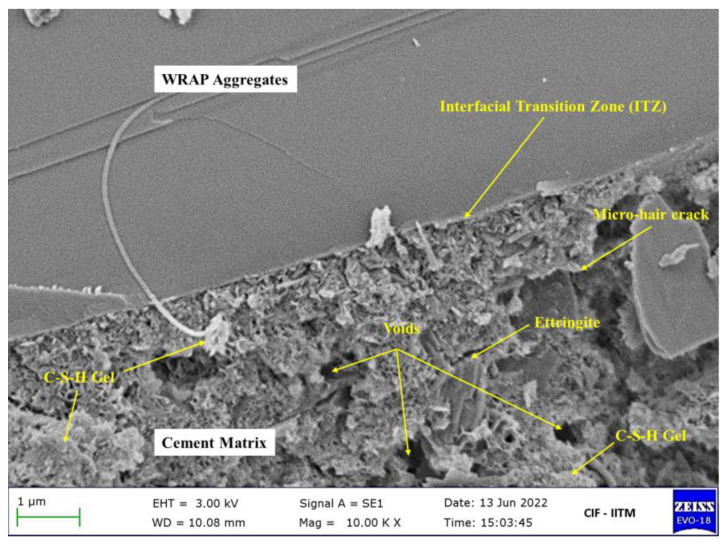
Microstructure of RAP concrete mixes using FE-SEM images.

**Table 1 materials-16-01504-t001:** Physical properties of RAP aggregates.

Properties	NCA	NFA	WRAP	DRAP	Standard Recommendations
Specific gravity	2.8	2.7	2.4	2.49	IRC: 44-2007
Water absorption (%)	0.62	0.76	0.5	1.78	IS: 2386-III
Density (kg/m^3^)	1650.4	1560.2	1543.8	1395.2	IS: 2386-III
Crushing Value (%)	21.25	18.85	12.8	15	<30%
Loss Angeles Abrasion (%)	24.75	20.45	13.75	18.6	<30%
Impact value (%)	16.65	14.80	10.2	12.56	<30%
Asphalt Content (%)	0	0		2.16	ASTM D2172

Notes: IRC Indian Road Congres.

**Table 2 materials-16-01504-t002:** Chemical Composition of Portland cement and ZSF admixtures.

Composition	SiO_2_	CaO	Fe_2_O_3_	MgO	Al_2_O_3_	MnO	K_2_O	Na_2_O	TiO_2_	P_2_O_5_
OPC (%)	37.28	49.76	2.75	2.12	5.83	0.04	0.73	0.82	0.37	0.30
ZSF (%)	92.75	0.40	2.57	0.75	1.08	0.05	0.50	1.6	0.10	0.20

**Table 3 materials-16-01504-t003:** Mix proportions of ingredients considered in RAP mixes in kg/m^3^.

Mix ID	% RAP	Mix Type	NCA	NFA	Coarse DRAP	Coarse WRAP	Cement	ZSF	Water
Control Mix	0	REF CMSF05 CMSF10 CMSF15 CMSF20 CMSF25 CMSF30	1360.5	745	0 0 0 0 0 0 0	0 0 0 0 0 0 0	360 342 324 306 288 270 252	0 18 36 54 72 90 108	184.5
DRAP20 WRAP20	20	DRAP WRAP	1088.4	745	272.1 0	0 272.1	324	36	184.5
DRAP40 WRAP40	40	DRAP WRAP	816.3	745	544.2 0	0 544.2	324	36	184.5
DRAP60 WRAP60	60	DRAP WRAP	544.5	745	816 0	0 816	324	36	184.5
DRAP80 WRAP80	80	DRAP WRAP	272.1	745	1088.4 0	0 1088.4	324	36	184.5
DRAP100 WRAP100	100	DRAP WRAP	0	745	1360.5 0	0 1360.5	324	36	184.5

**Table 4 materials-16-01504-t004:** Mechanical properties of the Different Concrete Mixes in Presence of ZSF.

Mix Type	Compressive Strength		Split Tensile Strength	
	7 Days Strength	28 Days Strength	7 Days Strength	28 Days Strength
	(MPa ± SD)	(MPa ± SD)	(MPa ± SD)	(MPa ± SD )
RM	31.56 ± 0.34	46.16 ± 0.89	3.50 ± 0.19	4.47 ± 0.34
CMSF05	33.80 ± 0.51	49.62 ± 0.90	3.69 ± 0.25	4.65 ± 0.33
CMSF10	41.19 ± 0.27	52.70 ± 0.61	3.61 ± 0.32	4.78 ± 0.31
CMSF15	43.70 ± 0.73	54.55 ± 0.75	3.64 ± 0.53	4.85 ± 1.23
CMSF20	43.53 ± 0.38	54.49 ± 0.83	3.65 ± 1.02	4.84 ± 1.18
CMSF25	42.83 ± 0.93	54.02 ± 1.21	3.63 ± 0.91	4.83 ± 1.13
CMSF30	42.42 ± 0.42	53.33 ± 1.07	3.64 ± 0.94	4.83 ± 1.24

Notes: SD = standard deviation of two specimens tested, RM = Reference Mix.

**Table 5 materials-16-01504-t005:** Statistical analysis using Two-way analysis of variance (ANOVA).

Source of Variation	*p*-Value	F	Fcr	*p* < α	F > Fcr	Significant Difference
Density	0.00000127	16.31	4.21	True	True	Yes
Compressive Strength	0.000000825	4.32	3.84	True	True	Yes
Flexural strength	0.00000436	12.69	3.84	True	True	Yes
Split tensile strength	0.00000375	4.54	3.84	True	True	Yes

## Data Availability

Data can be provided upon request.

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
