# Peer review of "Feasibility Study of Reclaimed Asphalt Pavements (RAP) as Recycled Aggregates Used in Rigid Pavement Construction"

_materials, 2023, doi:10.3390/ma16041504_

Round 1

Reviewer 1 Report

This manuscript evaluates the Feasibility study of Reclaimed Asphalt Pavements (RAP) aggregates used in Rigid Pavement Construction. The manuscript requires revision, and the possibility of acceptance is assessed after revision.

(1). The summary is relatively complete on the whole, but in lines 15-17, "RAP aggregates were used in varienamounts in the current study, including 0%, 20%, 40%, 60%, 80%, and 100% replacements for natural aggregate in the concrete mix. The description is relatively complex, and it is recommended to modify it, given the dosage range of RAP materials.

(2). The background is relatively complete on the whole, but it focuses on the importance and necessity of RAP as aggregate, and there is only one sentence to summarize the determination of the content range. It is recommended to appropriately increase why 100% RAP is not selected as aggregate. At the same time, the first few sentences of the first section and the second section in the background are slightly repeated, please delete as appropriate.

(3). The line drawing in Figure 3 is a little messy, so it is recommended to redraw it. The picture should be simple and clear.

(4). It is recommended to add a more intuitive flow chart at the "5Testing Procedure".

(5). The test results shall be analyzed for reasons rather than simple data listing.

(6). In Figure 5, only the data of 7Days have error bars and the length of error bars are consistent. It is recommended to supplement the error bars of other data and check the original data for errors.

(7). Please supplement the error bars in Figure 6 -8.

(8). Please check Figure 11 and delete the red line under the word "Ettringite"

(9). The conclusion of the article should be an important summary of the research results rather than a simple summary of the test results. Please simplify the research conclusions.

Reviewer 2 Report

materials-2136027-peer-review-v1

Feasibility study of Reclaimed Asphalt Pavements (RAP) aggregates used in Rigid Pavement Construction

General Comment

The manuscript deals with the results of laboratory investigations to determine the optimum amount of Reclaimed Asphalt Pavement for manufacturing concrete mixes for concrete pavements. I recommend to publish this manuscript after minor revision, essentially related to typo corrections. 

Specific comments

My comments and suggestions are reported in the attached file.

Reviewer 3 Report

1. The abstract should be short and a snapshot of the research paper. Please revise the abstract. 

2. "as" word missing in the title.  The author should write Reclaimed Asphalt Pavement as a recycled aggregate.......

3.  Line 96-97 is completely wrong. This is not a new study. There are lots of studies that have already been done on this topic since 2000.  Please refer following research article. 

https://www.sciencedirect.com/science/article/abs/pii/S0713274300800240

4. Lots of typo errors i.e. lines 44,48,50, ..... and so on.  Please check the whole manuscript.

5. what is a new perception of your study? How your study differs from the previously published studies?

6.  Nothing is mentioned about the ANOVA method in the introduction part. 

7. Please remove unnecessary information from Lines 45-85.

8.  In Figure 2, What is "N"?  As you mentioned in the abstract NA is - Natural Aggregate but what is "N"?

9. What do you mean by Dirty RAP Aggregate - DRP? How does it differ from other aggregates (i.e. WRA and NA)?

10. Please add the chemical properties of all the raw materials.

11. You can add the pictorial view of Zirconia silica fumes (ZSF) so that readers can understand the form of this material.

12. Lots of errors were found in abbreviations. i.e. NA, NCA, N, NFA, WRAP. DRAP

13.  Please remove Figure 4. Also, the author should remove unnecessary figures throughout the manuscript. 

13.  Figure 5 indicates the information about Compressive Strength in RAP-based Concrete. but the title of section 6.1 is  Variation of Compressive Strength on DRAP and WRAP aggregates.

Where are the results of WRAP aggregate compressive strength?

Have you compared the RAP and WRAP aggregate results?

14. " WRAP based aggregate" This word is wrong. Please remove this word.  Also Please write RAP recycled aggregate instead of RAP aggregate.

15. Please mention whether you have to use the RAP recycled aggregate as Coarse aggregate or Size.

16.  nothing is mentioned about the effect of ZSF on compressive strength.

17. here, there are two varying parameters i.e. RAP recycled aggregate and ZSF. Please add individual results for both the varying parameters.

18. Figure 5 shows the merely minor reduction as a replacement for RAP recycled aggregate by NA.  Please mention the main reason and justify your findings with previous research.

19.  Similar comments for sections 6.2 and 6.3 as mentioned above in comment no.13.

20. authors have just written increase/decrease strength or so on. But they haven't justified their findings. 

21. In Figure 9, Why does the trend of RAP 20 differ from the other mix?

22. Statistical Analysis part is too short. Please revise it.

23.  Please write the Rank for all varying parameters in ANOVA analysis.

24. Conclusion part should revise.  In the conclusion part, Lines 344, 345 

"admixtures ZSF act as supplementary cementitious material to increase the workability and ......."

Here authors have mentioned workability but there are no results or tests available on workability in the result and discussion chapter.

No correlation was observed between the title, abstract, result and discussion, and conclusion. 

Round 2

Reviewer 1 Report

The authors have revised this manuscript carefully.

Reviewer 3 Report

The authors have revised the manuscript as per the comments.